# Access to Vaccines in Floodplains and Hard-to-Reach Areas of the Brazilian Amazon: The Contribution of Street-Level Bureaucrats and the Use of Social Technologies

**DOI:** 10.3390/ijerph22050680

**Published:** 2025-04-25

**Authors:** Jair Araújo de Lima, Anízia Aguiar Neta, Suze Mary Camurça Assis, Bruno de Oliveira Rodrigues, Helena Ribeiro

**Affiliations:** 1Instituto de Filosofia, Ciências Humanas e Sociais, Universidade Federal do Amazonas (UFAM), Rua Rodrigo Octávio Jordão Ramos 3000, Manaus 69077-000, Brazil; jair.lima@ufam.edu.br (J.A.d.L.); brunorodrigues@ufam.edu.br (B.d.O.R.); 2Fiocruz—Fundação Oswaldo Cruz Amazônia, Rua Teresina 476, Manaus 69067-070, Brazil; anizia.neta@fiocruz.br; 3Program Imuniza SUS of the Secretaria Municipal de Saúde (SEMSA) in the Municipality Careiro da Várzea, Careiro da Várzea 69255-000, Brazil; suze.enf@gmail.com; 4Faculdade de Saúde Pública, Universidade de São Paulo (USP) Av. Dr. Arnaldo 715, São Paulo 01240-904, Brazil

**Keywords:** vaccine, Amazon, floodplains, Street Level Bureaucracy, social technologies

## Abstract

Introduction: Access to vaccines provided by the Brazilian National Immunization Program (NIP) to populations living in floodplains and hard-to-reach areas of the Amazon is complex and conditioned by the geographic characteristics of the region. The success of vaccination campaigns requires different strategies, technologies, and the involvement of professionals whose work goes beyond standard procedures and vaccination protocols. Objectives: To investigate the specificities of the immunization process of populations inhabiting floodplains and areas of difficult access in the municipality of Careiro da Várzea, in the state of Amazonas, Brazil. To analyze the theoretical and practical aspects of the National Immunization Program in the region. Methods: The case study included qualitative-descriptive techniques that combined data analysis, document analysis, and participant observation to reveal different socio-sanitary aspects of the immunization process of the Amazonian populations studied. The concepts of Street Level Bureaucracy and Social Technologies guided the analysis and description of the immunization process in the area studied. Results: The study described the geographic conditions, the social technologies used, and the individuals involved in the immunization process of the populations of communities and villages in flooded areas of the Amazon in Brazil. The high temperatures in the region create the need for thermal control in the storage of vaccines during their transfer to the communities and villages. The local coordination of the Imuniza SUS Program acts as a strategic mediator between the different bodies, ensuring the population’s access to vaccines, which means that the actions of government agents (Street Level Bureaucracy) are crucial to the functioning of the immunization program. Conclusions: The success of the immunization campaigns in the hard-to-reach flooded areas of the municipality is due to the existence of a virtuous cycle arising from the synergy between the different stakeholders that make up the immunization service; there is a clear relationship between the vaccination coverage rates achieved and the municipal administration’s commitment to public health. The immunization rates achieved in the municipality studied were compatible with the average established by the Brazilian Ministry of Health. This case study might enhance knowledge about health practices in this important world region.

## 1. Introduction

The vaccination coverage of Amazonian populations is influenced by different factors. The socioeconomic level of the population is a determinant for their adherence to the immunization program, but the events supposedly attributable to vaccination or immunization (ESAVI) are also determinants, even more so when associated with low understanding of the risks related to diseases, a reality observed even in northern countries [1]. Factors such as misinformation, lack of planning by people and families, and the fear of suffering possible side effects caused by vaccines impact the vaccination coverage of Amazonian populations.

In addition to factors related to educational, cultural, and socioeconomic aspects, another factor that directly impacts the vaccination coverage of Amazonian populations is regional geography. The access of immunization teams to floodplain lands and hard-to-reach areas is complex, and its feasibility depends on the strategies adopted to reach the flooded lands. The actions of the National Immunization Program (NIP)—the national public policy that makes vaccines, serums, and immunoglobulins available to all regions of the country through the Unified Health System (SUS)—are planned by health professionals who act as street-level bureaucrats within the scope of the SUS Immunization Program, at the local level. These professionals make use of certain social technologies and are willing to extrapolate standard operating procedures and vaccination protocols and interact with users of the health system who are also the targets of immunization campaigns.

Brazil has had important achievements in the field of childhood immunization in the first year of life, resulting in significantly reduced morbidity and mortality rates in this age group. Over decades, the actions of the NIP have made it possible to eradicate smallpox, polio, hepatitis B, diphtheria, neonatal tetanus, yellow fever, and tuberculosis in its severe form [2]. However, when it comes to the immunization process of Amazonian populations, depending on the time of year, the vaccination schedule may suffer delays that, associated with numerous other factors characteristic of the region, can result in postponing the eradication of some diseases.

The Federal Government makes available, free of charge, in the NIP’s National Vaccination Calendar, 48 immunobiological agents, 31 vaccines, 13 serums, and 4 immunoglobulins. Vaccines include those indicated for groups with special clinical conditions, such as “people with HIV or individuals undergoing treatment for some diseases (cancer, kidney failure, among others), applied at the Reference Centers for Special Immunobiological (CRIE), and COVID-19 vaccines and others administered in specific situations” [3]. Of the 31 vaccines made available by the NIP, 12 make up the mandatory annual vaccination calendar of the municipality of Careiro da Várzea, the case studied, but all vaccines are available to the population under the SUS Immunization Program, whose teams ensure that they arrive at the destination in ideal conditions of conservation and effectiveness. The WHO establishes that vaccines should be stored in temperatures from +2 °C a +8 °C at the Health Centers and during transportation, which should not take more than 48 h [4]. The Brazilian Ministry of Health adopts these same parameters of positive (+2 °C a +8 °C) and negative (−25 °C a −15 °C) temperatures in the cold chain for vaccine storage in short and long terms, and during transportation. Temperatures are inspected in the vaccines’ rooms and at the administration points [5].

Although the NIP has offered a great contribution to raising the health levels of the Brazilian population over the years, studies show that the average maternal mortality rate in Amazonian municipalities is three times higher than the Brazilian average. This reality is even worse in areas of difficult access and floodplain areas, where access to health services is hampered by isolation, low population density, and precarious and insufficient means of transportation, factors that mean the states of the Legal Amazon have the worst health conditions in the country [6].

The geographical singularities of the Amazon region certainly interfere in the organization and supply of Primary Health Care (PHC) services [7,8]. The establishment of an exclusive format in the provision of health services for the riverine areas of the Amazon brought visibility so that more resources from the federal government were allocated to health care services for these populations [8], favoring the delivery of vaccines in floodplains and areas of difficult access. In theory, more resources mean improvements in the structure of human resources and transport in the region.

The provision of financial resources for health takes place through the management pact between the federal, state, and municipal levels, a financing dynamic that guarantees the transfer of funds for the maintenance and permanence of priority programs within the scope of PHC, including the SUS Immunization Program. The River Family Health (eSFF) teams working in the Basic Health Units (UBSF) are responsible for ensuring the population’s access to vaccines in floodplains and hard-to-reach areas of the Amazon. In these areas, immunization is affected by factors such as the distance between the localities/villages/communities and the municipal headquarters that serves them, the structure of the river health units available in each municipality, and the awareness of users regarding the willingness to adhere to the vaccination process.

The objective of this case study was to investigate the specificities of the immunization process of populations inhabiting floodplains and areas of difficult access in the municipality of Careiro da Várzea, in the state of Amazonas, Brazil, detailing the theoretical and practical aspects of the National Immunization Program. To this end, the geographic singularities of the region, the strategies adopted by the health professionals who make up the immunization teams, understood here as street-level bureaucrats, and the fundamental social technologies used in the implementation of this public policy are detailed, explained, and analyzed. Those elements, practices, and stakeholders are common to the floodplains and hard-to-reach areas of the immense Amazon region. So, this case study might enhance knowledge about health practices in this important world region, as well as in remote areas in other regions and countries.

## 2. Methods

This study combined the techniques of document analysis and analysis of secondary data. Studies based on secondary data do not require submission to the Research Ethics Committee, according to Resolution 466/12 and 510/16 of the Ministry of Health [9,10].

Secondary data were collected from vaccination protocols, vaccination maps, e-SUS spreadsheets and data panels, and newsletters from the Ministry of Health of Brazil (BRASIL, Ministério da Saúde, Vacinação, available at https://infoms.saude.gov.br/extensions/SEIDIGI_DEMAS_VACINACAO_CALENDARIO_NACIONAL_COBERTURA_RESIDENCIA/SEIDIGI_DEMAS_VACINACAO_CALENDARIO_NACIONAL_COBERTURA_RESIDENCIA.html [11] (accessed on 12 January 2025); BRASIL, Ministério da Saúde, Coortes Vacinais—Papilomavírus Humano (HPV), accessible at https://infoms.saude.gov.br/extensions/SEIDIGI_DEMAS_VACINACAO_HPV/SEIDIGI_DEMAS_VACINACAO_HPV.html [12]) (accessed on 12 January 2024).

Data were analyzed though descriptive statistics, and vaccination percentages from the population were calculated based on population registered from previous census data. For Human Papillomavirus, data were analyzed by age groups.

Furthermore, it employed participant observation, which made it possible to detail the dynamics of the immunization process in the area studied.

Participatory research consists of a modality of qualitative social research that necessarily implies the participation of the researcher in the context, group, or culture in which he or she performs his or her research, and his/her insertion with the subjects who are involved in the process. Participatory research, therefore, takes place in a “natural environment,” that is, without the need for artificially prepared environments or situations. In the case of this study, participant observation was conducted by the author/nurse who coordinates the NIP in the municipality of Careiro da Várzea, who kept different records of the phenomenon studied through technical notes, photographs, videos, and audio recordings in which she described the situations considered relevant, which places her in the process as a researcher-informant. The other authors accessed the information collected in the field and worked on the organization, analysis, and interpretation of these data, attributing meaning to them and transforming them into scientific knowledge [13]. The mixed methods aimed to verify if the tailored interventions implemented in the area reflected in effective vaccination coverage.

The concepts of Street Level Bureaucracy and Social Technologies guided the approach to the theme precisely because they contemplate the importance of contact between public agents and citizens/users of public policies, which should result in effective deliveries to the population, a practice identified in the performance of health professionals who operate the immunization process of populations in floodplains and areas of difficult access in the locations studied.

## 3. Results

### 3.1. Sociodemographic and Economic Aspects of the Study Area: Careiro da Várzea

Careiro da Várzea is an urban municipality located in a floodplain area at the merging of the Negro and Solimões rivers, as its name implies. The floodplain environments of the Amazon estuary are areas that are periodically flooded due to changes in the rainfall regime of the region and are subject to variations in the level of the rivers, twice a day and seasonally [14]. The Negro and Solimões rivers, which bathe the Amazon region, have their flood period between the months of January and June, which is when the water overflow invades and floods in the marginal areas occurs. These marginal areas are the floodplains, so-called because they are periodically flooded by the waters of rivers, lakes, streams, and *paranás* [15], and they are where the communities and villages contemplated in this study are located.

The municipality of Careiro da Várzea is one of the twelve that are part of the health region called “Manaus, Entorno e Alto Rio Negro,” and is located at an approximate distance of 30 km from Manaus, the capital of the state of Amazonas, in a straight line, by river. Its territorial area is 2627.474 km^2^ [16], and its population is 19,637 inhabitants, according to the 2022 Census of the Brazilian Institute of Geography and Statistics (IBGE) [17]. The municipality occupies the 37th position in the population index classification of municipalities in the state of Amazonas [18]. Its population is organized into 105 communities and 11 villages, and the demographic density is 7.47 inhabitants per km^2^. The GDP per capita is BRL 13,140.09 (USD 2161 in January 2025), being the 4th highest in the health region and the 22nd in the state of Amazonas [18]. The Human Development Index (HDI) of the municipality is 0.569, considered low.

The main economic activity of the municipality is agriculture, with pineapple being the most commercialized item; cattle raising is the second main economic activity in the municipality. The period of river flooding brings difficulties to the movement of people and compromises the performance of routine activities, but, at the same time, favors the supply of fish in the communities, lowering the price of fish in these areas [19]. Among the sectors that bring together the largest number of workers are public administration, defense and social security, water transport, and retail trade [19].

During the period of river flooding, the communities and village populations usually migrate from one location to another in search of protection against isolation and difficulties in mobility and access to services, food, and medicines. In periods of extreme flooding, houses built on stilts tend to be completely submerged, causing the total loss of furniture and equipment for domestic use. This mobility of people between the communities in the region during the flood period of the rivers is clearly perceived in the routine activities of health professionals, as there is a certain variation in the care data recorded in the e-SUS-AB System (SUS System in Primary Care), often causing inaccuracies in vaccination coverage data, for example, and in the municipality’s health indicators.

### 3.2. Sanitary Structure of Careiro da Várzea

The public health services of the municipality of Careiro da Várzea are structured as follows: a Health Secretariat; a Basic Floating Health Unit in the municipal headquarters; fifteen Basic Family Health Units (between fixed and mobile) that serve the population of communities and villages far from the headquarters; a Health Surveillance unit; a Primary Care Pharmacy; and a State Hospital Unit, which is currently managed by the municipal sphere.

### 3.3. The Logistics of Immunization in the Communities and Villages of Careiro da Várzea

The immunization of the population of Careiro da Várzea is guaranteed by the National Immunization Program (NIP), of the Ministry of Health (MS), and conducted locally by health professionals from the *Immuniza SUS* Program. The types and quantities of vaccines designated both for the population of the Careiro da Várzea headquarters and for the communities and villages that are far from the central area (commonly called the municipal headquarters) are sent by the State Health Department located in the capital Manaus. The logistics of transport to the seat of the municipality consists of travel by two routes, one by road and the other by river. The vaccines are transported by road to the port of the Product Supply Center (CEASA), located in the south of Manaus, and then they are sent to Careiro da Várzea, a journey of approximately 40 min by boat crossing the Solimões River. Upon arrival in the municipality, the packaging of the vaccines is under the care of the Municipal Health Department, where there is a vertical refrigerator with a temperature that is maintained between 2 °C and 8 °C. This equipment is also known as a cold room or vaccine refrigerator, and it ensures that the vaccines are properly preserved until the moment of their administration. Monitoring their temperature is the responsibility of a laboratory technician. Because the municipality has 95% of its area submerged during the annual flood period, access by the Negro and Solimões rivers is often unfeasible. In these cases, it is necessary to travel through the border municipality of Autazes, by the Gurupá *Paraná* (stream); this is a safer route in times of large flooding, although the voyage takes longer, turning the maintenance of adequate temperature into a challenge.

The municipality has ten Family Health Strategy teams, who are responsible for organizing and coordinating the ten existing vaccination rooms. The mobile units work in the format of the River Basic Health Unit (UBSF), structured to float on the river. Some of them were originally boats that have been adapted to enable health actions in floodplains and areas of difficult access. In some communities, in addition to the UBSF, a fixed Basic Health Unit (UBS) is maintained.

The SUS Immunization Program currently serves 116 locations in the municipality of Careiro da Várzea, 105 communities and 11 villages, a very complex service when considering the difficulties of access and the low demographic density of the region. The distribution of vaccine rooms is organized as shown in Table 1.

The number of vaccines made available is calculated based on data from the population registered in the e-SUS, in order to ensure the vaccination coverage of the population, and its distribution is made among the 10 vaccine rooms belonging to the municipality (1 room in the municipal headquarters and 9 in communities and villages). Each vaccination room has the permanent presence of a nurse and a nursing assistant, and these teams provide care to the communities and their surroundings, as shown in Figure 1.

When the COVID-19 pandemic challenged the organization and capacity of public health services at the global level, health teams working in the Amazon were already used to clearing the forest, facing the flood and drought of the rivers in small motorboats known in the region as “rabetas” and “voadeiras”, or in larger and faster motorboats, such as the so-called “jet boats” used to travel longer distances. However, coping with the difficulties produced by the geographical characteristics of the floodplain areas is quite complex, and the health actions and policies currently being developed in the region are far from contemplating the totality of the health demands of its population. In this regard, the municipality of Careiro da Várzea “faces numerous difficulties, and has a huge need for effective, integrative and equitable public health policies, as well as professional-user interaction to achieve the qualification of self-care, promoting health and preventing diseases” [20] p. 3.

Despite the historical underfunding of the SUS, the management of immunization campaigns in floodplains and hard-to-reach areas of the Amazon takes into account—not only during public health emergencies, but in the daily life of health services—the health situation in the context of each action, in addition to the financial costs involved in the process. The calculation of these costs includes transportation and the time of accommodation and stay of the immunization teams in the municipalities, communities, and villages, as well as the time and cost of getting the vaccines to all locations without changes or damage to their immunizing capacity. The goal, in each campaign, is to vaccinate the largest number of people in the shortest possible time, and the established goal is always to vaccinate 100% of the population.

After the vaccines arrive at the municipal headquarters, the teams meet to analyze the map drawn for the immunization action or campaign. This is the moment when managers distribute professionals by teams, having recent weather information as an essential factor. Thus, they decide on the access routes to communities and villages, personal protective equipment, and adequate transportation. The materials on the boats are organized by the crew members themselves, usually a helmsman (locally called a boatman) and the helmsman’s assistant, with the help of the health team, which remains attentive to the thermal packaging of the vaccines and the safety of the professionals on board. The journey to the communities can take between 1 and 3 h in normal traffic, but it also varies depending on the difficulties encountered in each situation and the water levels of the rivers. Access to the indigenous villages of the Murutinga Pole can take much longer, as the route requires entering areas where the forest is denser and surrounded by lakes and *igapós* of the Amazon.

Major health actions take place in the Amazonian municipalities once a year and involve the participation of complete teams of the Family Health Strategy: doctors, nurses, dentists, nutritionists, social workers, veterinarians, nursing technicians, and community health agents who work in the area. On these occasions, the team of the River Basic Health Unit acts in the identification of possible extraordinary situations in which its intervention is required, either to solve an immediate local problem or to provide the referral of users for procedures at a level of greater complexity within the health system. An example of this situation is the occurrence of snakebites, which are very common in floodplains during the river flooding period. In these emergency cases, the user receives first aid at the UBSF itself and is then referred to a reference hospital in the capital.

In the Amazon, navigation professionals generally have extensive experience, which does not mean that there are no moments of tension during the journey to the most distant locations. It is not uncommon for the “boatmen” to have to untangle boats that are stuck to some roots at the riverbed. There are also situations in which boats need to travel very slowly, depending on the movement of other vessels along the waterways. The arrival time in communities and villages also takes into account the need for refueling. The river gas stations that serve the needs of the boats also have the function of preventing the illegal transportation of fuel on board the vessels.

In situations of major river floods, the municipal headquarters of Careiro da Várzea can be completely flooded, making access to villages and communities very difficult for health teams. The collapse of small wooden bridges that connect the riverside to certain localities is common. These occurrences make it impossible for both health teams and residents of the flooded areas to travel safely, requiring creativity and determination from the teams to reach their destinations. It is necessary to find a way to communicate what happened to the municipal headquarters, making all the protocol records, but it is not always possible to wait for the bridges to be repaired or rebuilt, which can take months. A common situation is when masonry houses and stilts are almost completely submerged, requiring the use of small canoes to move within the urban area of Careiro da Várzea; the streets become riverbeds in floodplain areas, and nothing can be done except wait for the water level to drop.

The health authorities of the municipalities monitor the evolution of the floods with the available meteorological services, a resource also limited by the difficulty of connecting to the internet network in these areas during periods of major river floods. In 2019, for example, Careiro da Várzea had 100% of its municipal area submerged by the flooding of the Rio Negro, making the road that leads to the municipality impassable and making it difficult to access food, inputs, medicines, and services, especially in the villages and communities farthest from the municipal headquarters. In situations like this, the maintenance of living and health conditions is affected and is possible only through cooperation with the state level. Creativity and partnerships are needed in order to accomplish the mandatory annual vaccination calendar (Figure 2).

The volume of water in floodplains tends to decrease at a slow pace when the rains begin to cease. At this stage, the waters can be pooled along the urban and rural perimeters for several days, giving rise to other sanitary problems. A classic example is the proliferation of the Aedes Aegypti mosquito, which has caused the infection and even death of many people in the Amazon over the years.

In February 2024, the NIP started to make dengue vaccines available in regions of Brazil where the disease is prevalent, but that year, the administration of the vaccine was released only for the child and adolescent population. In the municipality of Careiro da Várzea, especially in the communities and villages, it was initially observed that people were hesitant to receive a new vaccine, the possible reactions of which were not yet fully known, a fact also captured by the participant observation technique.

The Health Overview of the Municipality of Careiro da Várzea recorded the prevalence of Human Papillomavirus in adolescents aged between 13 and 17 years who were living without specialized medical follow-up, which could mean “lack of information […] as to the form of prevention of the disease, which is worrying because it is a major risk factor for cervical cancer and can be prevented by a vaccine offered by the SUS” [20] p. 7. In this sense, it is important to highlight the need for more—or more comprehensive—health education actions in the communities and villages of the municipality, aiming to make known the HPV vaccine, made available by the NIP for children and adolescents between the ages of 9 and 14 years. Vaccination coverage data for the period of 2019 to 2024 are presented in Table 2 and Table 3.

In cases where vaccination coverage exceeds 100%, the following situations might have occurred: (1) The number of people vaccinated in a given area exceeded the expectations created from the number of people registered in the e-SUS System. In this case, it is found that human mobility in the region—due to the conditions of isolation that cause difficulties in accessing services, food, medicines, etc.—influenced the vaccination process, generating the overcoming of coverage. In the case of Table 2, for example, it is possible—and this occurs frequently in Amazonas—that female adolescents have sought vaccination care in a municipality other than the one in which they originated in the e-SUS, causing the percentage of vaccination in the municipality where the service was provided to be higher than expected based on the System’s records. Or: (2) The registration data of the e-SUS System did not reflect the projections of the municipality’s population growth, a common fact in the intervals between the Demographic Census compiled by the IBGE every ten years due to high population mobility. It is from the data generated by the Demographic Census that public health policies are established, and it is, therefore, unlikely that annual records reflect, in 100% of cases, accuracy in the relationship between the data registered in the public systems and the population living in the municipalities. The excess of 100% means that more persons than expected got vaccinated.

Table 4 shows the vaccination coverage obtained in the administration of mandatory vaccines in the municipality of Careiro da Várzea between 2019 and 2024. Because some vaccines are administered at the same time, vaccination coverage reaches the same percentage, as is the case of Meningococcal and Hepatitis B, whose coverage level was 121.94% in 2019; the same case occurs with the Hepatitis B and Penta Valent vaccines in the years 2020, 2021, 2022, 2023, and 2024.

The Hepatitis B vaccine in newborns aged 30 days or less was the one with the lowest number of doses administered throughout the period, and this is due to the fact that there is no maternity hospital in the municipality of Careiro da Várzea, leading pregnant women to travel to the maternity hospitals in the capital, where newborns receive the first doses of mandatory vaccines. In adulthood, Hepatitis B reached coverage of more than 100% in the years 2020, 2023, and 2024. The BCG vaccine for TB, made available to children preferably up to 12 h after birth, also obtained low vaccination coverage in the period from 2019 to 2024, confirming vaccination in maternity hospitals in the state capital.

The vaccine against COVID-19 was widely administered throughout the state of Amazonas after its distribution by the Ministry of Health, but in floodplains and areas of difficult access, immunization teams faced the difficulties imposed by regional singularities. In the municipality of Careiro da Várzea, the National Health Data Network [21] registered, until 11 January 2025, 12,975 doses administered (2nd dose), which represents 67.36% of the coverage of the entire municipality, communities, and villages, and 8717 doses (3rd dose), which is equivalent to 42.45% of the vaccination coverage of the population in this area.

## 4. Discussion

### Street Level Bureaucracy and Social Technologies in the Vaccination Process of Populations in Floodplain Areas and Hard-to-Reach Areas

The concept of *Street Level Bureaucracy* was the basis for describing how public health agents interacted with people from the communities during the implementation of the NIP in the floodplains and areas of difficult access located within the geographic perimeter of the Amazonian municipality of Careiro da Várzea. Associated with this concept, we sought to understand the ways in which social technologies compose the process, considering that geographical characteristics require health teams to appropriate specific technologies, in whose execution the participation of the community becomes indispensable.

The description of the specificities of the immunization process of the populations of these areas brought to the professionals involved in this study the challenge of developing the ability to observe life in the Amazonian environment in order to perceive its peculiarities and, at the same time, interpret whether the professional modes and practices adopted by the health teams favor the effectiveness of the ongoing actions/campaigns. Thus, the actions of the teams are centered on people/users/citizens, the final customers of health services, and on geographical factors very much dependent on the dynamics of the waters.

The participant observation process revealed that, within the scope of the National Immunization Program (NIP), the importance of the performance of street-level bureaucrats is undeniable. And the fact that the NIP is “an integral part of the World Health Organization Program, with partnerships established with UNICEF and contributions from Rotary International and the United Nations Development Program (UNDP)” [3], makes it a privileged locus for the action of these bureaucrats, as the interinstitutional dimension and international cooperation expand the possibilities and favor protagonist action in the generation of assertive deliveries to society, while ratifying the need for its presence in the community.

Public policies are established by different individuals, groups, or organizations that play a role in the political arena. The “individuals” are, objectively, the stakeholders who influence the political process and exercise behaviors and interests that may vary, depending on the roles they play in the political scenario [22]. See the section on stakeholders’ roles in Appendix A.

The implementation stage is, among the stages of the public policy cycle, the one that puts the policy into practice. Therefore, it is a complex process that involves people, power, needs, disputes, resources, skills, knowledge, and limits [23]. Such complexity has led [24] to consider that the implementation of public policies is not a neutral process, since the procedures and interactions inherent to their implementation influence their content and results.

There has been an increase in studies on Street Level Bureaucracy in Brazil due to the interest of scholars in the processes of public policy implementation [22]. The authors point out that since the first decade of the 2000s, there has been an expansion of systematic studies on the implementation of these policies in different topics of academic interest.

Street Level Bureaucracy is one of the levels of public administration bureaucracy [25]. The role of street bureaucrats consists of direct interaction with citizens/users of a given public service; therefore, they represent the link between the state and society. Street-level bureaucrats offer an innovation to traditional health policy mainly because they directly affect the governability and governance of the political-administrative system of any government. This means that these stakeholders are able to capture local needs and transform them into part of the public policy agenda, as well as to influence the process thanks to their freedom to make decisions (discretion). Such dynamics are brought about by mediations between elements of the local contexts of action vis-à-vis the normative structure of the government’s public policy [26,27,28]. The decisions that these bureaucrats need to make in the implementation of public policies are permeated by uncertainties and pressures inherent to the process [29]. Such situations require great analytical skills, a challenge for street bureaucrats whose involvement with the community, coupled with important technical skills, enables them to propose effective solutions to the problems identified.

The specificities of the role played by street-level bureaucrats enable them to lead certain transformations in implementing public policies, since each reality and each situation is subject to the context and social structure, among other factors, requiring the ability to act in immediate, personal, empathetic, direct, and impactful ways in the lives of citizens/users [29], while at the same time, maintaining the necessary distance to ensure technical and professional character. In this sense, refs. [30,31] explain that acting to meet such needs and expectations requires a great capacity for organization and categorization of demands.

Maynard-Moody and Musheno [32] analyzed the performance of street-level bureaucrats through participant observation, among other techniques, and came to the conclusion that decisions are made considering the specificities of each case, based on the participation of peers and the moral, social, and cultural values observed in the relationships established with the users of the policies during the process of implementation.

The field of public health finds increasing utility in Street Level Bureaucracy, considering the level of societal complexification and the demand for a growing modernization of the state’s structure of action, through the implementation of public policies that respond to socio-sanitary demands. This fact re-signifies the role of street bureaucrats as it opens space for the perception of new arrangements and other functions within different contexts, making it possible to contemplate the diversity of modern societies [28].

Ref. [30] observed a particular community insertion when studying the performance of Family Health Program community health agents, as an example of the application of Street Level Bureaucracy in the field of public health. Another study in this same field of knowledge was conducted by Lima and D’Ascenzi [33], who sought to apprehend, through qualitative research, the influence of the discretion of street bureaucracy on the National Humanization Policy in basic health units in Porto Alegre.

Street-level bureaucrats have a direct role in the decisions made institutionally around the NIP. These individuals are considered central to understanding the process, its implementation, and achieving the expected results [31]. In this sense, the effectiveness of the NIP is directly related to the performance of street-level bureaucrats and their ability to mobilize people, institutions, and powers to implement the steps related to immunization everywhere in the Brazilian territory.

In the Amazon—where hydrography is a determining factor in access to vaccines when it comes to floodplains and hard-to-reach areas—it is necessary for health professionals to establish different strategies and to join forces with other stakeholders in the target community to overcome local regional barriers. In the municipality of Careiro da Várzea, its communities, and villages, the work of these street-level bureaucrats takes place throughout the process of immunizing the population. From the moment communication about the vaccination campaign begins within communities and villages—organizing and structuring actions, involving and mobilizing people—until the completion of the vaccination process, it is possible to see their influence with the community. They are the ones who activate the key people in the communities, who, in turn, enable the waterway access of the boats that contain the vaccines, sometimes stagnant near the communities due to weather conditions. In the localities, negotiations between street-level bureaucrats and community members often guarantee certain logistical arrangements without which it would be impossible to perform certain health actions. The community, through key people, also mobilizes the necessary communication and transportation to ensure the population’s attendance at the vaccination rooms during the campaigns, positively impacting vaccination coverage in areas where access is most difficult.

The dynamics that make the population of floodplains access vaccines can vary in the extension of the Brazilian Amazon. Several factors may interfere in this process, including the distance between the municipality and the state capital; the types of access, whether road, waterway, or air (in some locations in the Amazon, access involves the use of three types of vehicles—cars, boats, and airplanes); the structure of health services and the organizational capacity of the regional health departments; the number of immunization team members in each municipality; and the alignment of street-level bureaucrats with key people in the assisted community, among other factors.

The success of the work of street-level bureaucrats is directly linked to their ability to create and/or influence the use of social technologies that the process of making vaccines and immunizers accessible to the population of floodplains and hard-to-reach areas of the Brazilian Amazon requires. This process can be observed in situations where creativity and inventiveness are required from immunization agents.

The concept of social technologies is based on human actions that aim to transform nature with the aim of improving society’s living conditions, a process in which humanity establishes a relationship of struggle and opposition with nature, but at the same time, of complementation and cooperation [34]. Social technologies are a way of “designing, developing, implementing and managing technologies aimed at solving social and environmental problems, generating social and economic dynamics of social inclusion and sustainable development” [35] p. 2.

An important characteristic of social technologies is the ability to include community individuals who, while participating in their construction, are benefiting from them. In this regard, the simple replication restricted to a Social Technology (ST) can result in reducing the degree of autonomy and organization of its future beneficiaries, unless it involves them in processes of resignification of the problem and the technology that needs to be (re)appropriated by the social group [36].

Social technologies are interactive, associative tools developed with the participation of the community and are powerful in generating effective inclusive solutions that, when shared and collegial, have the potential to transform social reality. Another characteristic of social technologies is the ability to bring together multidisciplinary groups focused on problem-solving [36].

The creation, development, implementation, and management of social technologies make use of creativity, inventiveness, and other skills and resources with the central objective of responding to the demands of the users of a given public service. This process can only be successful if there is multidisciplinary capacity to immerse itself in reality, in interaction with the target users of the policy or public action. Thus, the search for a user-centered solution adopts a participatory and transdisciplinary posture “synthesizing technological, economic, strategic, ethical, social and environmental values […] capable of reconciling the interests of innovation actors and protagonist users” [36] p. 3.

The immunization actions taken by the NIP in the floodplains and hard-to-reach areas of the municipality studied place the users of this health service at the center of decision-making, previously identifying, through the application of a situational diagnosis, the weaknesses and threats to the process, as well as the potentialities and opportunities available, increasing the reach of vaccination coverage.

It is inappropriate that important tools used to enable the vaccination of populations in floodplains and hard-to-reach areas of the Amazon are treated as arrangements created to help solve certain obstacles identified in the process. In practice, the interactions between different stakeholders, knowledge, and skills promote experiences that, when executed time and time again, are improved, making it possible to design a technological orientation to create effective solutions built socially, at the local level.

The vaccination process implemented in the floodplains and hard-to-reach areas evidenced the existence of different social technologies whose validation must be attested by health professionals working in these areas. The theorists who conceptualize social technologies, some of which are mentioned above, affirm the importance of such technologies for the immunization process, and clarity about them is fundamental for their validation, registration, and evolution in order to contribute to overcoming important challenges to the health of Amazonian populations.

The analysis of the data referring to the health context of the municipality studied also showed that there are certain discontinuities in the execution of health programs, which are caused by political changes in the scope of management. Such discontinuities impact the effectiveness and guarantee of the achievement of all the established immunization goals.

The NIP’s investments in the disease prevention of the Brazilian population are increasing every year, but the immunization process must consider the emergence of new infectious diseases that, associated with the social diversity and geographical characteristics of a country of continental dimensions such as Brazil, challenge the immunization policy and, consequently, the health managers in the public spheres.

The prevalence of dengue in the municipality of Careiro da Várzea, and in its communities and villages, indicates the increasing need for water monitoring in floodplain areas, regardless of the time of year, whether in the dry or rainy season, since the waters that supply the population depend on rainfall, causing changes in its components and affecting the health of people and animals.

The study focused on a single municipality, Careiro da Várzea and its surrounding areas, which may limit the generalizability of the findings. In spite of this limitation, we believe and recognize that similar situations occur in the immense Amazonian floodplain.

## 5. Final Remarks

Documentary and data analysis made it possible to understand the dynamics of the immunization process in the communities and villages of the Amazonian municipality of Careiro da Várzea. This analysis, associated with the concept of Street Level Bureaucracy, made it possible to attribute great relevance to the participation of street-level bureaucrats in the process of immunizing the population. The participant observation technique made it possible to obtain, intentionally and systematically, information and data on the necessary actions, protocol or not, of the immunization process of the populations that inhabit areas of difficult access marked by the predominance of flooded spaces typical of floodplains.

The performance of public health agents (street-level bureaucrats) is, therefore, decisive to make the mandatory annual vaccination calendar viable. These professionals have great influence with local individuals, making them essential for reaching health system users and obtaining the agreements and participation necessary to achieve satisfactory levels of problem-solving capacity in the SUS.

Different techniques for collecting and analyzing qualitative data, combined with the presentation and analysis of vaccination coverage data from the Ministry of Health, provided clarity and reliability to the process studied.

The adoption of social technologies in the implementation of the immunization policy was considered very relevant, as it involves elements, practices, and common people to the floodplains and hard-to-reach areas of the immense Amazon region. The immunization demands a multiplicity of actions, with equally diverse methods and approaches, depending on the geographic location within the Brazilian territory, and its ethnic and cultural diversity.

The local coordination of the National Immunization Program acts as a strategic mediator between the different instances to ensure people’s access to vaccines at the farthest locations in adequate programed time, no matter how difficult the path or how sparse the population. The success of immunization campaigns in floodplain areas and hard-to-reach areas of the municipality of Careiro da Várzea is due to the existence of a virtuous cycle arising from the synergy between the different individuals who make up the immunization service. There is a clear relationship between the vaccination coverage achieved and the commitment of the municipal management to public health, making the immunization rates achieved compatible with the average established by the Ministry of Health.

## Figures and Tables

**Figure 1 ijerph-22-00680-f001:**
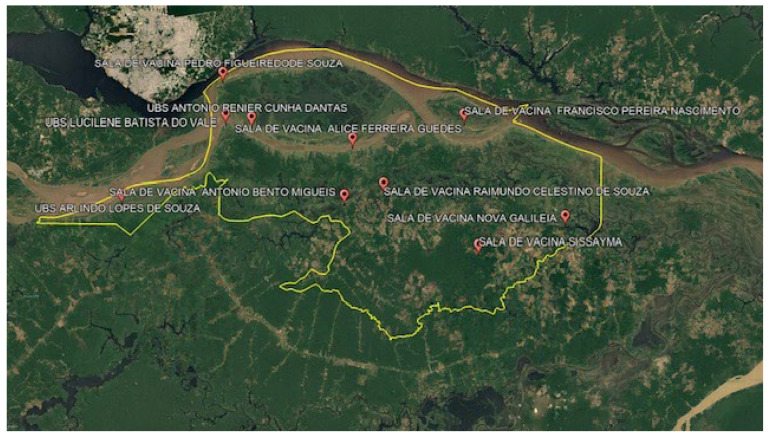
Aerial image with the location of the 9 vaccination rooms in operation within the scope of the NIP in communities and villages in the municipality of Careiro da Várzea. Source: Image Google Earth georeferenced by the authors.

**Figure 2 ijerph-22-00680-f002:**
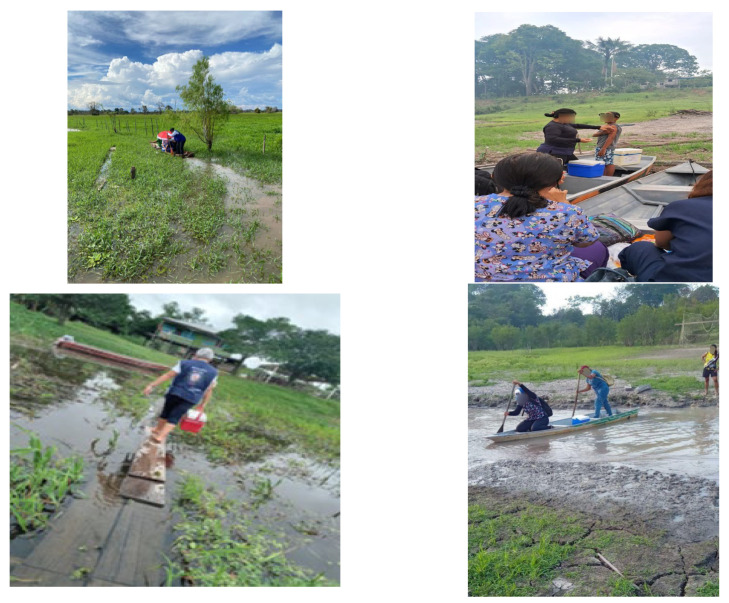
Vaccine transport and vaccination in floodplains and hard-to-reach areas in Careiro da Várzea. Source: Photos taken by the authors.

**Table 1 ijerph-22-00680-t001:** The vaccination rooms in activity in floodplain areas and areas of difficult access in the municipality of Careiro da Várzea, type of Health Unit (whether fixed and/or mobile), number of communities and villages served, and types of transportation used for access.

Name of the Vaccination Room	Type of Unit (Fixed/Mobile)	Number of Communities and Villages	Transportation Route
Dr. Antônio Renier	Fixed	5 communities	Road and river
Lucilene Batista	Fixed	2 communities	Road and river
Alice F. Guedes	Mobile and fixed	18 communities	River
Pedro Figueiredo	Mobile and fixed	11 communities	River
Arlindo Lopes	Mobile	14 communities	River
Francisca Pereira Nascimento	Mobile	17 communities	River
Antônio Bento Migueis	Fixed	11 communities	Road and river
Raimunda Celestino de Souza	Mobile and fixed	18 communities	River
Nova Galileia	Mobile	9 communities	River
Polo Murutinga	Mobile	11 Indian villages	River
Total		116	

Source: Organized by authors based on data from e-SUS of 2024.

**Table 2 ijerph-22-00680-t002:** Vaccination coverage (single dose) in females, aged between 9 and 14 years, referring to the Human Papillomavirus (HPV), in the municipality of Careiro da Várzea and its communities and villages, in the period between 2019 and 2024.

Age Group	% 2019	% 2020	% 2021	% 2022	% 2023	% 2024
9 years	77.84	30.81	24.86	40.54	72.97	67.03
10 years	100.63	138.36	57.23	59.75	86.79	105.03
11 years	79.39	106.06	142.42	65.45	69.09	90.30
12 years	90.76	75.00	97.83	132.07	62.50	61.96
13 years	209.04	92.02	74.47	97.87	133.51	62.23
14 years	186.98	233.14	102.96	84.02	110.06	149.70
9 to 14 years	88.00	110.19	82.48	80.67	89.43	98.39

Source: Organized by the authors based on data from Sistema e-SUS/MS.

**Table 3 ijerph-22-00680-t003:** Vaccination coverage (single dose) in males, aged between 9 and 14 years, referring to the Human Papillomavirus (HPV), in the municipality of Careiro da Várzea and its communities and villages, in the period between 2019 and 2024.

Age Group	% 2019	% 2020	% 2021	% 2022	% 2023	% 2024
9 years	1.16	0.58	0.58	12.21	44.77	51.16
10 years	5.77	1.44	1.44	9.13	44.71	58.17
11 years	80.00	41.05	33.68	40.53	67.89	73.16
12 years	99.38	133.95	67.28	64.20	86.42	96.91
13 years	87.69	93.33	115.90	62.56	65.64	76.92
14 years	92.11	96.32	99.47	122.11	72.63	71.05
9 to14 years	60.25	59.44	53.00	51.48	63.12	70.73

Source: Organized by the authors based on data from Sistema e-SUS/MS.

**Table 4 ijerph-22-00680-t004:** Vaccination coverage of mandatory vaccines in the municipality of Careiro da Várzea, its communities, and villages in the period between the years 2019 and 2024.

Vaccines	2019	2020	2021	2022	2023	2024
Hepatitis B (≤30 days of life)	18.14%	30.38%	38.89%	38.42%	26.67%	17.65%
TB–BCG	19.83%	35.02%	41.92%	49.47%	27.62%	20.59%
Rota Virus	118.99%	94.51%	120.2%	120%	66.19%	91.18%
Meningococcal	121.94%	95.78%	150%	128.95%	65.71%	35.29%
Hepatitis B	121.94%	87.34%	131.82%	128.42%	70%	70.59%
Penta Valente	121.52%	87.34%	131.82%	128.42%	70%	70.59%
Yellow Fever	94.94%	68.35%	81.31%	101.05%	52.38%	100%
Hepatitis A	148.95%	86.92%	121.63%	131.05%	76.67%	132.45%
Pneumo 10	120.68%	70.69%	130.58%	133.16%	70.48%	97.06%
Polio VIP	123.21%	80.59%	153.35%	120%	72.38%	67.85%
Triple Viral 1st dose	128.69%	95.78%	141.41%	148.42%	79.09%	144.12%
Triple Viral 2nd dose	162.03%	59.92%	59.60%	106.84%	30.48%	105.88%

Source: Organized by the authors based on data from *Sistema e-SUS/MS*.

## Data Availability

The original contributions presented in this study are included in the article and Appendix A. Further inquiries can be directed to the corresponding author.

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
