# Peer review of "Access to Vaccines in Floodplains and Hard-to-Reach Areas of the Brazilian Amazon: The Contribution of Street-Level Bureaucrats and the Use of Social Technologies"

_ijerph, 2025, doi:10.3390/ijerph22050680_

Round 1
Reviewer 1 Report
Comments and Suggestions for Authors
Thank you for your submission on a very interesting topic! I have a few suggestions for your manuscript to consider:
Abstract:
Under Results and Conclusions, could a better term other than “actors” be used to describe the individuals involved in the immunization process. You could just use the word “individuals”.
Introduction: Do you have a reference for page 2 lines 52 63? It is unclear if it is still Macedo.
Page 2 line 63 Define “social actors”- are these the same as street bureaucrats?
Page 3 lines 99-109. Do you have a reference for this information? Is it Sousa 2023?
Objective: Please consider a different term than “actors”
Results:
4.3 The logistics of immunizations.. page 4 line 195. Should that abbreviation be NPI? The National Immunization Program was identified on page 2 line 56 as NPI.
Page 5 line 205. Should “cold season” be “cold chain” ? Please provide some context on how the cold chain is preserved? Do the containers have temperature monitors? How often are the temperatures recorded to ensure there are no temperature excursions.
Page 6 line 262. What is meant by “and others”- are these other logistical issues?
Page 7 line 287- consider removing the word “fuel” since you already have the word “refueling”
Page 8 line 323- please add the word “Human” to the papillomavirus so it reads Human Papillomavirus
Page 8 HPV vaccines tables: Are these tables documentation of all single doses of the HPV vaccine or completed series of the vaccines. Was there data on how many completed the HPV series and how many were partials? That might be interesting to see how many girls and boys completed the series.
Discussion:
Page 11 line 454 and 463- please consider using the word individuals instead of actors.
Page 11 line 465- Please confirm the correct abbreviation for National Immunization Program as NPI and not PNI
Page 12- there appears to be a lot of direct quotes consider paraphrasing these so there are not as many direct quotes
Page 12 line 532-533 what is the reference for this quote?
Page 12 line 534 and page 13 line 574 consider a different term rather than “actors”
References need to be translated into English please.
This was a very interesting project and I think would be valuable knowledge to the readers.
Author Response
REVIEWER 1
Abstract:
Under Results and Conclusions, could a better term other than “actors” be used to describe the individuals involved in the immunization process. You could just use the word “individuals”.
Change was made in the text.
Introduction: Do you have a reference for page 2 lines 52 63? It is unclear if it is still Macedo.
The paragraph is made according to the participant observation made by the authors. There is no reference.
Page 2 line 63 Define “social actors”- are these the same as street bureaucrats?
The term was substituted for stakeholders. They are not only the street bureaucrats. A table of functions was included as supplementary material.
Page 3 lines 99-109. Do you have a reference for this information? Is it Sousa 2023?
This information was obtained in the fieldwork.
Objective: Please consider a different term than “actors”
The term was substituted for individuals and stakeholders.
Results:
4.3 The logistics of immunizations.. page 4 line 195. Should that abbreviation be NPI? The National Immunization Program was identified on page 2 line 56 as NPI.
Throughout the paper the abbreviation was changed to NIP from National Immunization Program.
Page 5 line 205. Should “cold season” be “cold chain”? Please provide some context on how the cold chain is preserved? Do the containers have temperature monitors? How often are the temperatures recorded to ensure there are no temperature excursions.
Sorry for this mistake. It is cold chain. Addicional information was added.
Page 6 line 262. What is meant by “and others”- are these other logistical issues?
We excluded the word.
Page 7 line 287- consider removing the word “fuel” since you already have the word “refueling”
The word fuel was removed.
Page 8 line 323- please add the word “Human” to the papillomavirus so it reads Human Papillomavirus
Correction was made in the text and the word Human added.
Page 8 HPV vaccines tables: Are these tables documentation of all single doses of the HPV vaccine or completed series of the vaccines. Was there data on how many completed the HPV series and how many were partials? That might be interesting to see how many girls and boys completed the series.
Single dose was added in the title, as the information regarding completed series was not available in the Ministry of Health data.
Discussion:
Page 11 line 454 and 463- please consider using the word individuals instead of actors.
The word “actors” was substituted in all the manuscript.
Page 11 line 465- Please confirm the correct abbreviation for National Immunization Program as NPI and not PNI
Change made to NIP.
Page 12- there appears to be a lot of direct quotes consider paraphrasing these so there are not as many direct quotes
Text was changed and direct citations were paraphrased.
Page 12 line 532-533 what is the reference for this quote?
This is not a quote. We removed italics.
Page 12 line 534 and page 13 line 574 consider a different term rather than “actors”
The word was substituted for individuals.
References need to be translated into English please.
References were adapted to the format required by the jornal.
Reviewer 2 Report
Comments and Suggestions for Authors
Reviewer suggested comments mentioned below:
The revised manuscript still needs to be thoroughly rerevised and it was found that in many places grammatical errors and written sentences are long which is not clear what the authors want to say.
Mentioned references must be rechecked in between the text that is appropriate for sentences.
In the manuscript, the authors should revise the abstract. The abstract objective section should be revised and the written sentence can be divided into two sentences rather than one long sentence. Need to improvise sentences in the written abstract. It is not up to the publication mark.
In the introduction section authors write too much about vaccination policies and other things so can be shortened unnecessarily mentioned information. The introduction section can be improvised and only the accurate published literature information regarding the study is enough to cover in this part.
The objective should be written in the last paragraph of the introduction section rather than mentioned in a separate sub-title.
The methods section very poorly describes the sources of collected data in the study. It must be needs to provide more information about the study data collection methods including references from where data was collected and how much those sources are valid as a reference.
The manuscript arrangement is not fine as according to article. The written results information does not match to mentioned title and what the authors want to describe in this section. Its looks like a review article where the authors mentioned the geographical distribution of the location and mentioned health care difficulties and other things. The authors mentioned about only one main focus area which is Careiro da Várzea.
Line 154, 155: “Waters” should be rechecked, is it grammatically correct?
Comments on the Quality of English LanguageMust be improvise the English language.
Author Response
REVIEWER 2
The revised manuscript still needs to be thoroughly rerevised and it was found that in many places grammatical errors and written sentences are long which is not clear what the authors want to say.
The manuscript was totlly revised and phrases were shortened and corrected.
Mentioned references must be rechecked in between the text that is appropriate for sentences.
All the references were rechecked.
In the manuscript, the authors should revise the abstract. The abstract objective section should be revised and the written sentence can be divided into two sentences rather than one long sentence. Need to improvise sentences in the written abstract. It is not up to the publication mark.
The abstract was improved.
In the introduction section authors write too much about vaccination policies and other things so can be shortened unnecessarily mentioned information. The introduction section can be improvised and only the accurate published literature information regarding the study is enough to cover in this part.
The authors considered importante to describe the vaccination policies of Brazil as they differ from other countries.
The objective should be written in the last paragraph of the introduction section rather than mentioned in a separate sub-title.
We included the objectives in the last paragraph of the introduction, as suggested.
The methods section very poorly describes the sources of collected data in the study. It must be needs to provide more information about the study data collection methods including references from where data was collected and how much those sources are valid as a reference.
Souces of collected data were included as well as the methods and official references.
The manuscript arrangement is not fine as according to article. The written results information does not match to mentioned title and what the authors want to describe in this section. Its looks like a review article where the authors mentioned the geographical distribution of the location and mentioned health care difficulties and other things. The authors mentioned about only one main focus area which is Careiro da Várzea.
The manuscript described that Careiro da Várzea as a case study of an Amazonian municipality where access to vacines is a big challenge due to its island location, intense floods and sparse population. However its example might elucidate similar situations in this immense region with an área largest than most countries in the world.
Line 154, 155: “Waters” should be rechecked, is it grammatically correct?
The term waters is very Much used in portuguese for the Amazon region, as there are many types of waters (brown, black, pristine, salty, from rivers, from stream (igarapés) or lakes, etc. The English Cambridge dictionnaire includes the word Waters.
Must be improvise the English language.
I believe the revisor meant to improve the English language instead of improvise. We have reviewed and improved all the text with the help of a native speaker.
Reviewer 3 Report
Comments and Suggestions for Authors
I have carefully read and reviewed the paper "Access to Vaccines in Floodplains and Hard-to-Reach Areas of the Brazilian Amazon: The Contribution of Street-Level Bureaucrats and the Use of Social Technologies." The paper addresses a topic of significant academic and practical relevance, focusing on the challenges and strategies for implementing immunization programs in remote and challenging environments. The case study approach with qualitative-descriptive techniques provides valuable insights into the socio-sanitary aspects of the immunization process in the Amazon region. However, the paper requires revisions before it can be considered for publication. Below, I outline my key recommendations:
Major:
- While the paper introduces the concepts of "street-level bureaucracy" and "social technologies," the discussion of the theoretical framework is underdeveloped. The paper would benefit from explicitly linking these concepts to established frameworks in public health or policy implementation. This will better highlight the study’s theoretical contributions and position it within the broader literature.
- The study focuses on a single municipality (Careiro da Várzea) and its surrounding areas, which may limit the generalizability of the findings. The discussion section should address this limitation and consider the potential impact on the broader applicability of the results. Expanding the study to include more diverse locations within the Amazon region could strengthen the paper.
- Although the qualitative sampling is comprehensive, the integration of qualitative data with quantitative vaccination coverage data to derive key conclusions is not sufficiently clear. A more detailed explanation of how the two datasets support each other is needed to enhance the robustness of the findings.
- The recommendations in the conclusion are somewhat generic (e.g., "improving coordination among stakeholders" and "enhancing community engagement"). The authors should include more actionable suggestions, such as specific strategies for improving vaccine logistics, training programs for health workers, or innovative approaches to community mobilization.
- Some key arguments, such as those related to the logistical challenges of vaccine storage and transportation in remote areas, lack adequate citation. For example, the discussion of the impact of high temperatures on vaccine efficacy could benefit from additional references to WHO guidelines or related studies.
Minor:
6.Ensure consistent use of key terms throughout the paper. For example, the terms "street-level bureaucrats" and "public health agents" are used interchangeably, which can cause confusion. Defining terms clearly in the introduction and using them consistently will improve clarity.
7.The quality of the English language is generally good, but there are areas for improvement in terms of clarity and conciseness. Some sentences are overly complex and could be simplified for better readability. Additionally, minor grammatical and punctuation errors should be corrected.
Author Response
REVIEWER 3
While the paper introduces the concepts of "street-level bureaucracy" and "social technologies," the discussion of the theoretical framework is underdeveloped. The paper would benefit from explicitly linking these concepts to established frameworks in public health or policy implementation. This will better highlight the study’s theoretical contributions and position it within the broader literature.
In the topic discussion, the theoretical framework was better developped and linked to public health policy iplementation.
The study focuses on a single municipality (Careiro da Várzea) and its surrounding areas, which may limit the generalizability of the findings. The discussion section should address this limitation and consider the potential impact on the broader applicability of the results. Expanding the study to include more diverse locations within the Amazon region could strengthen the paper.
We included this limitation of a case study in the Amazon region in the discussion topic. However, the intention was to show that this situation is not unique in the whole of the region and it is a challenge to vaccinate people in áreas of sparse population with no roads, and only water transport.
Although the qualitative sampling is comprehensive, the integration of qualitative data with quantitative vaccination coverage data to derive key conclusions is not sufficiently clear. A more detailed explanation of how the two datasets support each other is needed to enhance the robustness of the findings.
The authors improved the discussion, showing that, in spite of the distance, the lack of infrastrure and resources, and the floods, the coverage of vaccination is similar to other areas of Brazil and the mandatory annual vaccination calendar is achieved in the municipality. Photographs were included in the article to illustrate the process.
The recommendations in the conclusion are somewhat generic (e.g., "improving coordination among stakeholders" and "enhancing community engagement"). The authors should include more actionable suggestions, such as specific strategies for improving vaccine logistics, training programs for health workers, or innovative approaches to community mobilization.
The authors included supplementary material, indicating the role of each stakeholder. In our view, the way the Community engagement is done is already successful.
Some key arguments, such as those related to the logistical challenges of vaccine storage and transportation in remote areas, lack adequate citation. For example, the discussion of the impact of high temperatures on vaccine efficacy could benefit from additional references to WHO guidelines or related studies.
We have included WHO guidelines for vaccine storage and transportation, as well as thos from the Ministry of Health.
Ensure consistent use of key terms throughout the paper. For example, the terms "street-level bureaucrats" and "public health agents" are used interchangeably, which can cause confusion. Defining terms clearly in the introduction and using them consistently will improve clarity.
We clarified the terms used as “street-level bureaucrats” in the discussion section. They can be the public health agents, which also have other functions. The table in the supplementary material describes the different functions of the professionals.
The quality of the English language is generally good, but there are areas for improvement in terms of clarity and conciseness. Some sentences are overly complex and could be simplified for better readability. Additionally, minor grammatical and punctuation errors should be corrected.
We have revised all the text, shortened some sentences for better readability and corrected some minor gramatical errors.
Reviewer 4 Report
Comments and Suggestions for Authors
To the authors: I am recommending against publication of this manuscript. You do a good job of showing some of the obstacles that you face in successfully delivering vaccines to your citizens living in the flood plains. I think that the sensitizing concept of "the street level bureacrat" is appropriate. However, you never really describe WHO served in this role-(Nurses, transporters, coordinators?). The specific adaptations that they made, also deserve more examples and description. The reason, I am recommending against a revision, however, is that your health data for showing the relative effectiveness of the campaign is out of date or flawed. Many of the cells in your tables exceed 100%. I could not meaningfully interpret these figures, and I am not sure how you can work around that in an article revision.
Best wishes
Author Response
REVIEWER 4
To the authors: I am recommending against publication of this manuscript. You do a good job of showing some of the obstacles that you face in successfully delivering vaccines to your citizens living in the flood plains. I think that the sensitizing concept of "the street level bureacrat" is appropriate. However, you never really describe WHO served in this role-(Nurses, transporters, coordinators?). The specific adaptations that they made, also deserve more examples and description. The reason, I am recommending against a revision, however, is that your health data for showing the relative effectiveness of the campaign is out of date or flawed. Many of the cells in your tables exceed 100%. I could not meaningfully interpret these figures, and I am not sure how you can work around that in an article revision.
We included fotos and better described the challenges regarding vaccination in those floodplains and remore áreas.
We improved the text where it is explained why some cells exceed 100% in the tables. In a country with strong population growth and internal migration it is common to have population projections that do not confirm data expected. Population counting is done by census every 10 years.
In cases where vaccination coverage exceeds 100%, the following situations might have occurred: 1) the number of people vaccinated in a given area exceeded the expectations created from the number of people registered in the e-SUS System. In this case, it is found that human mobility in the region – due to the conditions of isolation that causes difficulties in accessing services, food and medicines, etc. – influenced the vaccination process, generating the overcoming of coverage. In the case of Table 2, for example, it is possible – and this occurs frequently in Amazonas – that female adolescents have sought vaccination care in a municipality other than the one in which they originated in the e-SUS, causing the percentage of vaccination in the municipality where the service was provided to be higher than expected based on the System's records; or 2) the registration data of the e-SUS System did not reflect the projections of population growth of the municipality, a common fact in the interval between the Demographic Census carried out by the IBGE every ten years due to high population mobility. It is from the data generated by the Demographic Census that public health policies are established, and it is therefore unlikely that annual records reflect, in 100% of the cases, accuracy in the relationship between the data registered in the public systems and the population living in the municipalities. The excess of 100% means that more persons than expected got vaccinated.
Table 4 shows the vaccination coverage obtained in the administration of mandatory vaccines in the municipality of Careiro da Várzea in the period between 2019 and 2024. Because some vaccines are administered at the same time, vaccination coverage reaches the same percentage, as is the case of Meningococcal and Hepatitis B, whose coverage level was 121.94% in 2019; the same case occurs with the Hepatitis B and Penta Valent vaccines in the years 2020, 2021, 2022, 2023 and 2024.